# Microstructure and Property Evolution of Diamond/GaInSn Composites under Thermal Load and High Humidity

**DOI:** 10.3390/ma17051152

**Published:** 2024-03-01

**Authors:** Shijie Du, Hong Guo, Jie Zhang, Zhongnan Xie, Hui Yang, Nan Wu, Yulin Liu

**Affiliations:** 1State Key Laboratory of Nonferrous Metals and Processes, GRINM Group Co., Ltd., Beijing 100088, China; 2Institute for Advanced Materials and Technology, University of Science and Technology, Beijing 100083, China; 3GRIMAT Engineering Institute Co., Ltd., Beijing 101407, China; 4General Research Institute for Nonferrous Metals, Beijing 100088, China

**Keywords:** thermal interface material, reliability, temperature and humidity stress test, high-temperature storage, temperature cycling

## Abstract

As a thermal interface material, diamond/GaInSn composites have wide-ranging application prospects in the thermal management of chips. However, studies on systematic reliability that can guide the practical application of diamond/GaInSn in the high-temperature, high-temperature impact, or high-humidity service environments that are faced by chips remain lacking. In this study, the performance evolution of diamond/GaInSn was studied under high-temperature storage (150 °C), high- and low-temperature cycling (−50 °C to 125 °C), and high temperature and high humidity (85 °C and 85% humidity). The experimental results reveal the failure mechanism of semi-solid composites during high temperature oxidation. It is revealed that core oxidation is the key to the degradation of liquid metal composites’ properties under high-temperature storage and high- and low-temperature cycling conditions. Under the conditions of high temperature and high humidity, the failure of Ga-based liquid metal and its composite materials is significant. Therefore, the material should avoid high-temperature and high-humidity environments.

## 1. Introduction

In the thermal management of chip packaging, components such as high-thermal composite materials and heat pipes greatly improve the heat transfer speed and effectively reduce the chip’s temperature [1,2,3,4]. As an important heat dissipation component in thermal management systems, thermal interface materials (TIMs) assume the important function of reducing the thermal resistance of the contact interface. TIMs can effectively fill the gap between the heat source and the cooling component and reduce the service temperature of the semiconductor by lowering the thermal resistance that is generated by the air between the contact surfaces [5,6]. TIMs need not only excellent thermal conductivity but also electrical insulation, good elastoplasticity, appropriate fluidity and viscosity, a low thermal expansion coefficient, good cold and hot cycle stability, and wide applicability [7,8,9].

TIMs can be roughly divided into thermal gels, thermal greases, thermal gaskets, and phase change materials according to their properties [10,11,12,13]. Thermal gels and thermal greases are liquid thermal interface materials, which have the advantage of a low contact thermal resistance, and their failure forms are easy to dry and cake. The advantages of thermal shims and phase change materials are that they are easy to use, and their failure forms are mainly stress cracks. As paste thermal conductivity materials, diamond/GaInSn composites have the advantages of high thermal conductivity and high deformation; thus, they can be used in electronic packaging as high-performance TIMs [14]. Compared with the current commercial TIMs, namely, In and thermal grease, diamond/GaInSn composites have outstanding advantages and excellent properties, including high thermal conductivity, low interfacial thermal resistance, and high viscosity; however, their reliability is rarely studied [15,16,17].

The working environment of TIMs is consistent with that of chips. They face high-frequency vibration [18], thermal cycling [19], thermal shock [20], and high-humidity conditions [21], which can cause cracks, deformation, and other defects in TIMs, resulting in reduced heat dissipation efficiency and a decreased service life of chips. The optimal operating temperature of a chip is 70–80 °C, and a chip’s reliability decreases by 50% for every 10 °C increase in the temperature of a single electronic component [22]. Roy C K et al. [23,24] studied the thermal resistance of a Ga-based liquid metal after aging at 130 °C and cyclic heating and cooling at −40 °C to 80 °C. They found that the sample’s thermal resistance showed a negligible (<5%) decrease after long-term aging. This finding is instructive for our reliability study of diamond/GaInSn composites. Researchers tend to assume that the change in the sample’s thermal resistance comes from the reaction between the liquid metal and the packaging material [25]. This principle is reasonable but does not reveal the change in the TIM itself.

We use a semi-solid-state diamond/GaInSn composite material for thermal load and high humidity tests, mainly because this composite material has excellent comprehensive properties, such as high thermal conductivity, low thermal resistance, and appropriate viscosity and fluidity. It is also an excellent choice for high-performance TIMs. Therefore, this study aims to investigate the effect of thermal load and high humidity on the microstructure, the surface state, and the properties of diamond/GaInSn composites and their corrosion mechanism. Thus, this study can provide guidance for the future application of such materials.

## 2. Materials and Methods

### 2.1. Selection of Experimental Materials

The selected TIM is a diamond/GaInSn composite paste-like material. The scanning electron microscope (SEM) photo of its morphology is shown in Figure 1. SEM images were collected by using a HITACHI S4800 SEM at 20 kV (HITACHI, Tokyo, Japan). GaInSn alloy is a liquid metal at room temperature, and the alloy composition is the eutectic component of ternary alloy Ga_67_In_20.5_Sn_12.5_ [26]. Metal materials are purchased from Grinm Advanced Materials Co., Ltd. (Beijing, China). Eutectic GaInSn’s melting point is 9 °C. The diamond particles in the GaInSn/diamond composite material were selected in two sizes of 100 mesh (150 μm) and 500 mesh (13 μm), and the amount of the diamond that was added was 35 wt% [7]. The diamond purchased from Henan Zhongyuan Superhard Abrasiors Co., Ltd. (Luoyang, China). Dia/GaInSn composite material was prepared using the mechanical stirring method [27]. After being ultrasonically cleaned with alcohol, the diamond particles were mixed with GaInSn eutectic alloy by mechanical stirring to prepare a uniform diamond/GaInSn composite material. The specific measured performance parameters of liquid metals and composite materials are shown in Table 1.

### 2.2. Experimental Scheme

The experiments under a high-temperature (150 °C) storage environment, a high-temperature and low-temperature circulation (−50 °C to 125 °C) environment, and a high-temperature and high-humidity (85 °C, 85% relative humidity) environment were designed to explore the influence of the temperature and humidity environment on the thermal conductivity of GaInSn/diamond composite by measuring the aging and failure law of the composite.

The high-temperature storage aging test aimed to simulate the continuous high temperature experienced by the TIM [28]. The experiments followed the JESD22-A103 [29] standard. The experimental composite material was loaded into the Teflon crucible and placed into a blast air oven at a constant temperature. The temperature was set at 150 °C. Six aging cycles were set, including 0 (nonaging), 24, 48, 96, 240, and 480 h.

The temperature cycling (TC) tests aim to induce TIM failure/degradation associated with the cyclic changes in temperature during operation. Accelerated TC tests should adhere to a standard similar to JESD22-A104D [30] or JESD-A105C [31,32]. The temperature cycle aging experiment was conducted in the temperature cycle chamber. The time dependence of the temperature cycle is shown in Figure 2. Six aging period comparison groups were set to 0, 50, 100, 300, 500, and 1000 cycles. The morphology and the thermal conductivity of the aging test sample were tested. 

The high-temperature and high-humidity aging test box was used to test the failure of the material at high temperature and high humidity. The test conditions were 85 °C and 85% relative humidity [33,34,35]. Six experimental groups were set up: 0, 24, 48, 96, 240, and 480 h. The above aging test equipment was purchased from Guangdong LESTEST Co., Ltd. (Dongguan, China).

In the above experiments, three groups of parallel samples were set for each experimental group, with 20(±0.2) g for each group. The thermal conductivity test was conducted within 24 h after the samples were taken out of the test box. The test results were taken as the average of the three groups of parallel samples.

The density of the GaInSn composite TIM was measured via the drainage method. The specific heat capacity of the material was measured using a Differential Scanning Calorimetry (DSC, Netzsch 214Polyma, Netzsch, Selb, Germany) thermal analyzer, and the thermal diffusion coefficient was measured using the laser flash method (Netzsch LFA 467 HyperFlash). A 3D profilometer and scanning electron microscope (SEM HITACHI S4800) were used to characterize the surface morphology and its structure.

## 3. Results and Discussion

### 3.1. Experimental Analysis of the High-Temperature Storage Environment

Figure 3 shows the SEM morphology in the high-temperature storage experiment. The diamond in the GaInSn/diamond composite is an inorganic carbon material with poor wettability with a liquid metal. However, the diamond can form a “diamond–Ga_2_O_3_–liquid metal” wettability interface structure by adhering to the oxide of Ga on the surface as an auxiliary wetting method. In the preparation, the doped oxide Ga_2_O_3_ in GaInSn is used as the interface layer to assist wetting through the micro-oxidation of GaInSn. In the high-temperature storage experiment, the oxygen content in the composite material slowly increased with the extension of the high-temperature storage time. Thus, the oxide content in the liquid metal increased, thereby improving wettability, reducing the number of exposed diamond particles, and increasing liquid film folds on the liquid metal surface.

Due to the formation of an oxide film on the surface of the liquid phase, when the semi-solid composite was spread on the SEM sample table, the oxide film was subjected to tensile stress from the diamond, so the radial wrinkles centered around the diamond appeared on the sample surface [36]. The deeper the degree of oxidation was, the more obvious the film formation was, and the more wrinkles appeared (see Figure 3).

As shown in Figure 4, the performance of the material in the high-temperature environment from 24 h to 96 h was not only not reduced but abnormally increased. This phenomenon occurred because the preliminary oxidation generated a trace amount of gallium oxide in the high-temperature storage experiment. Moreover, the presence of gallium oxide promotes the wetting of the thermal conductivity-enhanced phase with the matrix. However, the material’s thermal conductivity is increased. A high-temperature storage time of more than 100 h deepens the material’s oxidation degree, changes the material’s surface from bright silver to dark gray, and decreases the composite’s thermal conductivity. Figure 4 shows that the material’s thermal conductivity begins to decline after 100 h with the extension of high-temperature storage time. Moreover, the material’s thermal conductivity tends to stabilize after 240–480 h of high-temperature storage.

Samples were taken from the surface and core of the tested material, respectively, and the oxygen content of the sample was measured using an Energy-Dispersive Spectrometer (EDS). Figure 5 shows that the oxygen content of the core did not change after 240 h of storage at a high temperature, but the oxygen content of the surface was increasing. The oxygen content of the material’s surface was higher than that of the material’s core, and the oxidation products were limited to the material’s surface, resulting in a small change in the material’s overall thermal conductivity. It can be seen that the liquid GaInSn phase on the surface of the composite reacted with oxygen at high temperatures, and the generated surface oxide protected the composite and prevented the further oxidation of the core material, so the oxygen content in the core remained unchanged. The thermal conductivity degradation of the material is also affected by the diamond particle size. The thermal conductivity of the diamond/GaInSn composite made with a 150 μm diamond particle size only decreased by 3% after 480 h of high-temperature storage. The thermal conductivity of the composite made with a 25 μm diamond particle size decreased by 7% after 480 h of high-temperature storage. The 25 μm diamond/GaInSn composite had a lower density and higher porosity than the 150 μm diamond/GaInSn composite, and the oxidation reaction spread more easily from the surface to the core. In the high-temperature storage environment, the high-temperature oxidation products oxidized easily on the surface and could not cause the deterioration of material properties. The key to the degradation of material properties lies in the oxidation inside the material.

### 3.2. Experimental Analysis of the High- and Low-Temperature Circulation Environment

In the experiment using high- and low-temperature cycling, the oxygen content in the composite increased with the increase in the number of cycles. Thus, the oxide content in the liquid metal increased, and its wettability was enhanced. We can observe the improvement in wettability using SEM, which demonstrates that the bare diamond particles are reduced, and the liquid film folds appear on the liquid metal surface (Figure 6a–e,g–k). However, excessive oxidation will cause a large number of lamellar gallium oxide to accumulate towards the interface [37], resulting in exposed diamond particles and destroying the wettability of the liquid phase and diamond (Figure 6f,l). Macroscopic photographs show that the entire surface of the material is tarnished. At the same time, the increase in oxygen content in the high- and low-temperature cycling was higher than that in the high-temperature storage experiment. This phenomenon shows not only wrinkles but also a scaly oxide film in the liquid phase, observed by SEM. The oxygen content of the experimental group with 100 cycles (250 h) was compared with that of the experimental group with 240 h high-temperature storage. The atomic ratio of the oxygen content after 100 cycles was 2.94%, which is slightly higher than the oxidation content of the experiment with high-temperature storage at the same duration.

The high- and low-temperature cycle aging pattern of the diamond/GaInSn composite is similar to the high-temperature storage aging pattern, as shown in Figure 7. At the beginning of the aging experiment, the material’s thermal conductivity increased, because the gallium oxide generated by oxidation promotes wetting. However, the material’s thermal conductivity did not show a stable trend after a long period of high- and low-temperature cycle aging. In particular, it consistently exhibited a decreasing trend. The thermal conductivity of the material decreased by 18.6% after 1000 cycles. Experimental studies indicate that the material undergoes a melting–crystallization–solidification–melting process in the high–low temperature cycle. The voids in the particle–liquid phase composite continue to move, disappear, and appear in the high–low temperature cycle because of the volume change in the material, which is caused by melting and solidification. As a result, the oxidation products that were originally concentrated on the surface spread to the core of the material, making the whole material fully oxidized (Figure 8). Thus, the material’s thermal conductivity continuously declined.

### 3.3. Experimental Analysis of the High-Temperature and High-Humidity Environment

Figure 9 shows the weight gain curve and corrosion rate curve of the diamond/GaInSn composite material at different times in a hot and humid environment. The mass of the composite material before the humid and thermal environment experiment was 20 g, as shown in Figure 9a. After the high-temperature and high-humidity experiment had been conducted for 480 h, the weight of the material increased by 4.9 g (150 μm diamond/GaInSn) and 5.8 g (25 μm diamond/GaInSn). At the beginning of the 24 h hot and humid environment experiment, uniform dark corrosion products appeared on the surface of the composite material, and these products were produced only on the material’s surface. Compared with the high-temperature storage experiment, the high-temperature and dry environment experiment did not lead to any change in the gloss of the material.

The distribution of corrosion products in the composite deepened. Moreover, the separation of the liquid metal matrix phase and the diamond reinforcement phase occurred during the 240 h test in a humid and hot environment. After the 240 h test in the hot and humid environment, the liquid metal content decreased, and most of the liquid phase reactions produced solid corrosion products. The macroscopic morphologies of the composite material at the same time were compared. The results show that the corrosion phenomenon of the composite material is more obvious in the humid and hot environment than in high-temperature storage conditions. Moreover, there were more corrosion products in the former than in the latter. The calculation of the corrosion rate shows (Figure 9b) that the corrosion rate of the material was lower than 50 g/m^2^h from 0 h to 96 h. The corrosion rate was the highest at 360 h and decreased after 360 h. In practical applications, the material fails when the solid–liquid phase of the composite material is separated and the material loses its paste shape, as shown in Figure 10. In the thermal conductivity test, the diamond/GaInSn composite failed at a longer duration than 240 h under hot and humid conditions. However, the diamond particles in the 25 μm diamond/GaInSn material cannot bear the effect of thermal enhancement after 48 h.

The diamond particles were extracted and screened after the wet and thermal environment experiments to explore the composition and the formation mechanism of the corrosion products. After the experiments, the composite materials were centrifuged and screened, and the diamond particles were selected for further analysis. Figure 11 shows the microscopic morphology of the surface of diamond particles in the 150 μm diamond/GaInSn composite and 13 μm diamond/composite after 240 h of humidity and heat. As shown in Figure 11a,d, many honeycomb-like structures were generated on the diamond surface. These structures are corrosion products, produced by composite materials in a hot and humid environment. Figure 11c,f show that the black contrast is diamond, and the gray contrast is a corrosion product. The diamond’s flat surface indicates that the corrosion product is not a reaction product of the diamond. However, the liquid metal adheres to the diamond surface after corrosion. The amount of corrosion products that combine on the diamond surface of different particle sizes is different. Most of the diamond crystal surface of the large-particle diamond is exposed. However, the surface of the small-particle diamond is completely wrapped by honeycomb corrosion products.

Figure 12 shows the EDS analysis of the diamond surface. The results show that the main chemical components of the corrosion products that are attached to the diamond surface are Ga and O. The contents of In and Sn are insignificant. However, the corrosion product is tightly bound to the surface of the diamond crystal but cannot uniformly cover the diamond surface. Thus, the EDS shows a large area of the exposed C signal. An XRD analysis of the screened diamond powder was performed. The results are shown in Figure 13. The results show that the main components of the honeycomb structure on the surface of the screened diamond particles are GaOOH and a small amount of SnO_2_. The main peak at 43.9° is consistent with the diffraction peak of the diamond (111) crystal face, and the main peak at 75.3° is consistent with the diffraction peak of the diamond (220) crystal face, so it can be concluded that the main component of the material is diamond. The diffraction peaks and other peaks at 33° indicate small amounts of SnO_2_ in the material. The other small diffraction peaks in the spectra are consistent with the diffraction spectra of GaOOH; in particular, the high 21°, 33°, and 37° spectral peaks are consistent with the (110), (130), and (111) crystal planes of GaOOH, and here, there is no correlation in the diffraction peak signals of Ga and Ga_2_O_3_ in the XRD pattern. This shows that the Ga and the oxide of Ga attached to the diamond surface are all transformed into GaOOH.

The failure mechanism of the composite in a hot and humid environment was further studied. The doped oxide powder in the liquid metal that was obtained after screening was collected. The solid-phase products were separated from the oxidation products and analyzed via EDS and XRD. The EDS analysis results of the corrosion products are shown in Figure 14. The figure shows that large particles of solid powder are generated by corrosion, and a small amount of GaInSn liquid metal is adhered to by the solid powder. The oxygen content in the liquid metal is minimal, and the main components of the solid particles are Ga and O. The XRD analysis of the corrosion product powder shows (Figure 15) that a small amount of liquid dopant exists in the powder, and a mantou peak can be found in the XRD spectrum. The solid crystal particles are GaOOH and a small amount of diamond. Based on the corrosion products, the chemical reactions resulting from the analysis are as follows:Ga^3+^ + 3OH^−^ ⟶ Ga(OH)_3_,
Ga(OH)_3_ ⟶ GaOOH + H_2_O.

When the liquid metal is immersed in the liquid metal droplet in the electrolyte solution, the surface of the liquid metal forms a net charge, and the reaction with water leads to the formation of a honeycomb crystalline layer on the surface of the diamond.

The reaction mechanism is shown in Figure 16. In the wet and thermal environment experiment, the Ga in the GaInSn liquid metal reacts with H_2_O in air under high-temperature and high-humidity conditions to generate Ga(OH)_2_, which is only confined to the composite surface at the beginning of the reaction. Thus, the reaction speed is slow. When the reaction occurs, only the corrosion products on the surface of the composite cause the bright silver diamond/GaInSn to lose luster. As the reaction progresses, Ga(OH)_2_ is enriched to the wetting interface of Dia and GaInSn and is dehydrated to form GaOOH. GaOOH and diamond are tightly combined to form a honeycomb structure. Thus, the wettability of diamond and GaInSn is seriously reduced, and the solid and liquid phases of the composite separate and overflow. Moreover, the H_2_O generated by hydrolysis reacts with Ga to continue to produce corrosion products. Thus, the diamond is connected through GaOOH, forming a complete solid-phase product. GaInSn reacts with water in the air to form GaOOH until the Ga is completely consumed.

Under the conditions of high temperature and high humidity, the failure of the Ga-based liquid metal and its composite materials is significant. Therefore, using the material in high-temperature and high-humidity environments should be avoided.

## 4. Conclusions

In the high-temperature (150 °C) storage environment, diamond/GaInSn reacts with oxygen in the air to form Ga_2_O_3_. Moreover, the material’s thermal conductivity improves in 24–96 h. The porosity of the 25 μm diamond composite is higher than that of the 150 μm diamond/GaInSn composite, and the oxidation reaction spreads more easily from the surface to the core. At 480 h, the 25 μm diamond composite’s thermal conductivity decreases by up to 7%. The key to the degradation of material properties lies in the oxidation that occurs inside the material.In the high- and low-temperature cycle experiment, the preliminary oxidation promotes the wetting of the thermal conductivity-enhanced phase and the matrix, and the material’s thermal conductivity is increased. The material undergoes a melting–crystallization–solidification–melting process. The voids in the particle–liquid phase composite continue to move, disappear, and appear in high- and low-temperature cycling. Thus, the oxidation products that originally concentrated on the surface spread to the inner part of the material, resulting in a continuous decrease in the material’s thermal conductivity.Under high-temperature and high-humidity conditions, the Ga_2_O_3_ in the GaInSn liquid metal reacts with H_2_O in the air to generate Ga(OH)_2_. Ga(OH)_2_ is enriched at the wetting interface between Dia and GaInSn and is dehydrated to generate GaOOH. GaOOH and diamond are tightly combined to form a honeycomb structure. Thus, the wettability of diamond and GaInSn is seriously reduced, and the solid and liquid phases of the composite separate. The liquid phase overflow leads to material failure.

## Figures and Tables

**Figure 1 materials-17-01152-f001:**
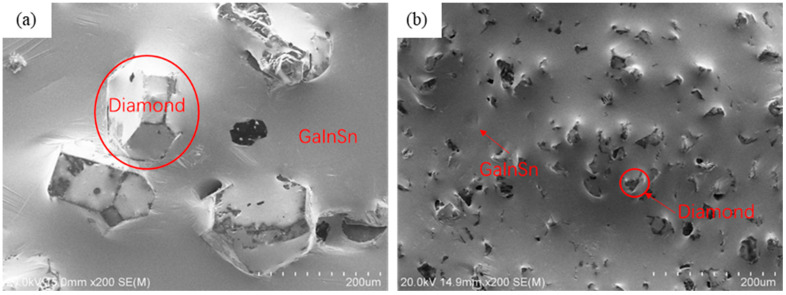
SEM photo of morphology of the diamond/GaInSn composite material: (**a**) 100 mesh (150 μm) diamond/GaInSn SEM; (**b**) 500 mesh (13 μm) diamond/GaInSn SEM.

**Figure 2 materials-17-01152-f002:**
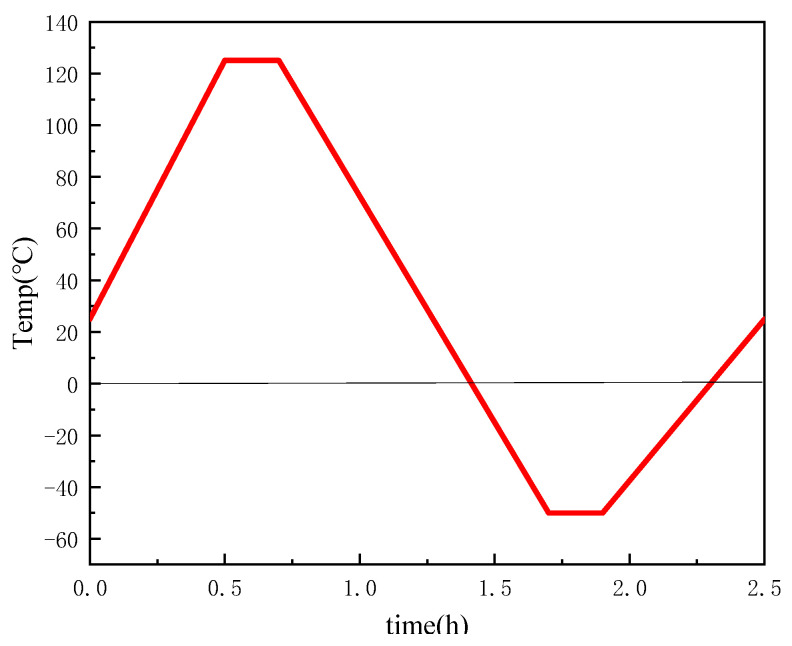
Time dependence of temperature cycle.

**Figure 3 materials-17-01152-f003:**
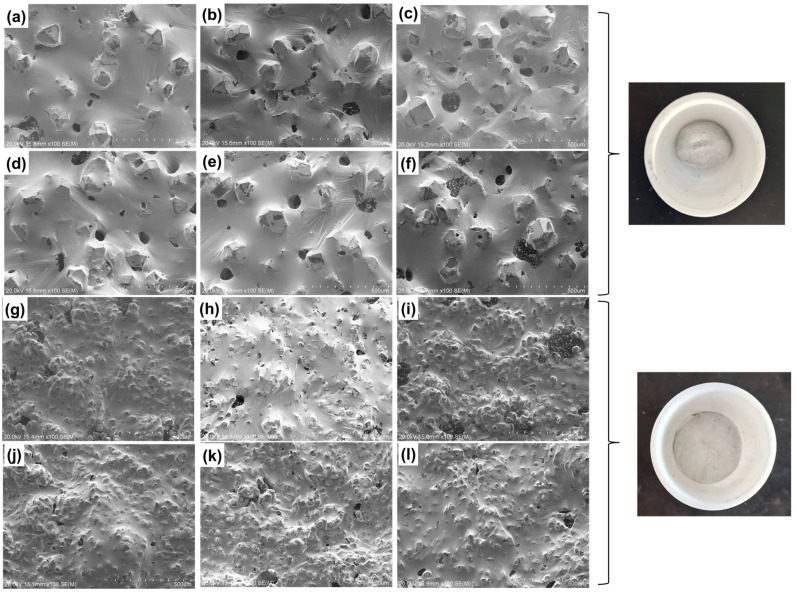
Morphology analysis of the high-temperature storage experiment. GaInSn/150 μm diamond: (**a**) 0 h; (**b**) 24 h; (**c**) 48 h; (**d**) 96 h; (**e**) 240 h; (**f**) 480 h. GaInSn/13 μm diamond: (**g**) 0 h; (**h**) 24 h; (**i**) 48 h; (**j**) 96 h; (**k**) 240 h; (**l**) 480 h.

**Figure 4 materials-17-01152-f004:**
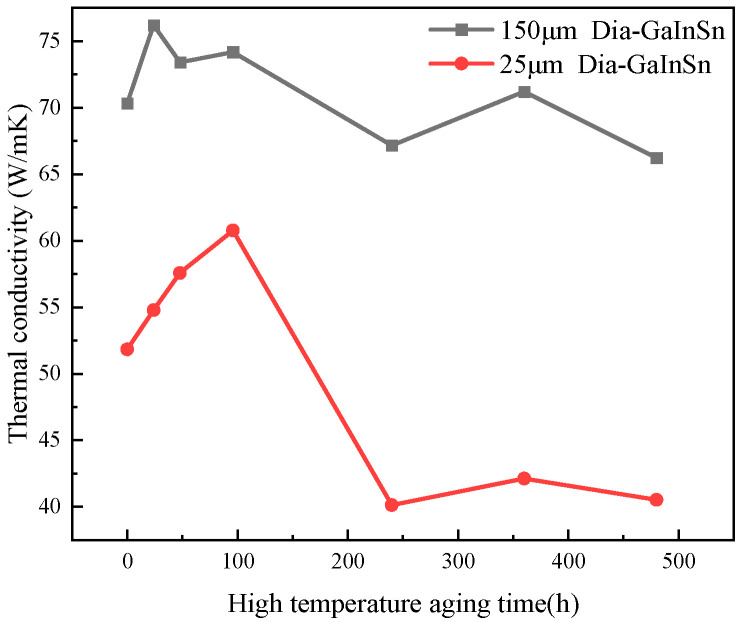
Experimental thermal performance in the high-temperature storage experiment.

**Figure 5 materials-17-01152-f005:**
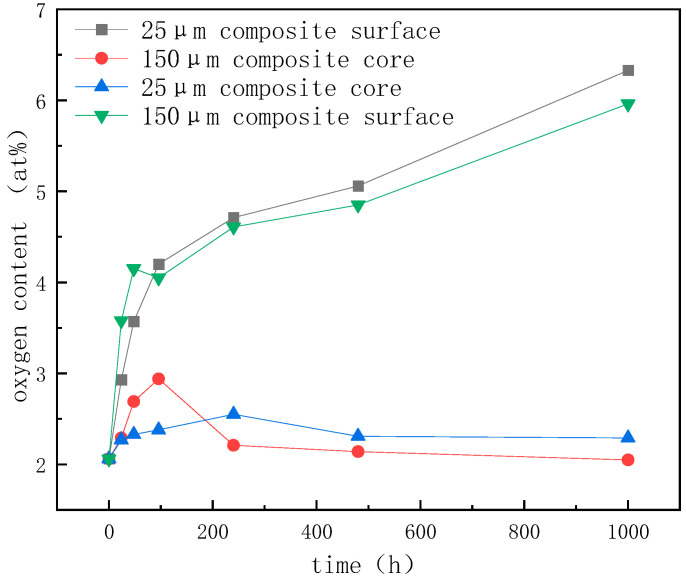
Oxygen content of materials analyzed via EDS.

**Figure 6 materials-17-01152-f006:**
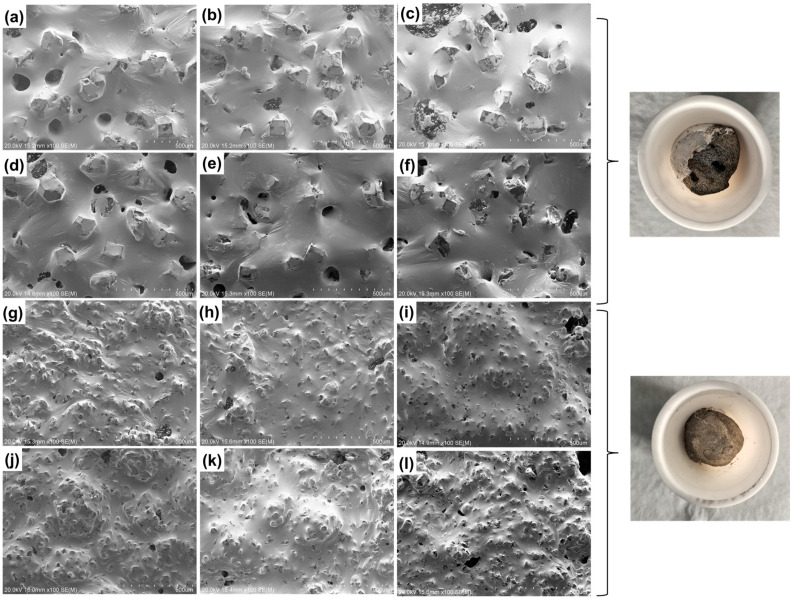
Morphology analysis of the high- and low-temperature cycle experiment. GaInSn/100 mesh diamond: (**a**) 0 times; (**b**) 50 times; (**c**) 100 times; (**d**) 300 times; (**e**) 500 times; (**f**) 1000 times. GaInSn/500 mesh diamond: (**g**) 0 times; (**h**) 50 times; (**i**) 100 times; (**j**) 300 times; (**k**) 500 times; (**l**) 1000 times.

**Figure 7 materials-17-01152-f007:**
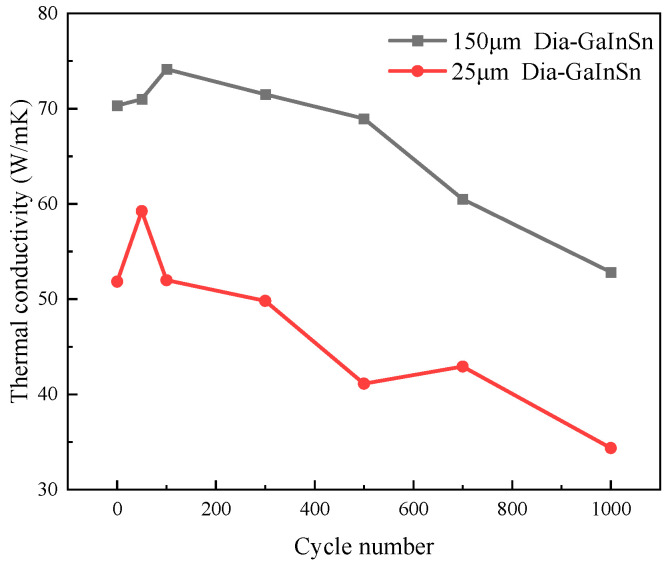
Thermal performance of the high- and low-temperature cycle experiment.

**Figure 8 materials-17-01152-f008:**
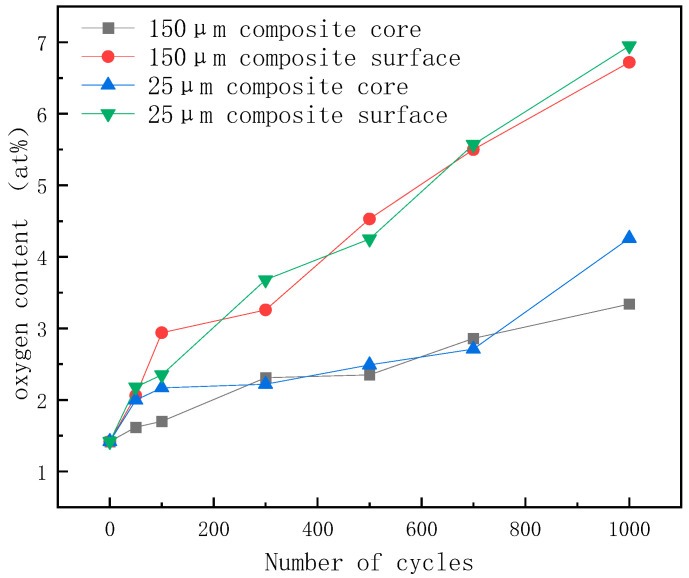
EDS analysis of material oxygen content after the high- and low-temperature cycling.

**Figure 9 materials-17-01152-f009:**
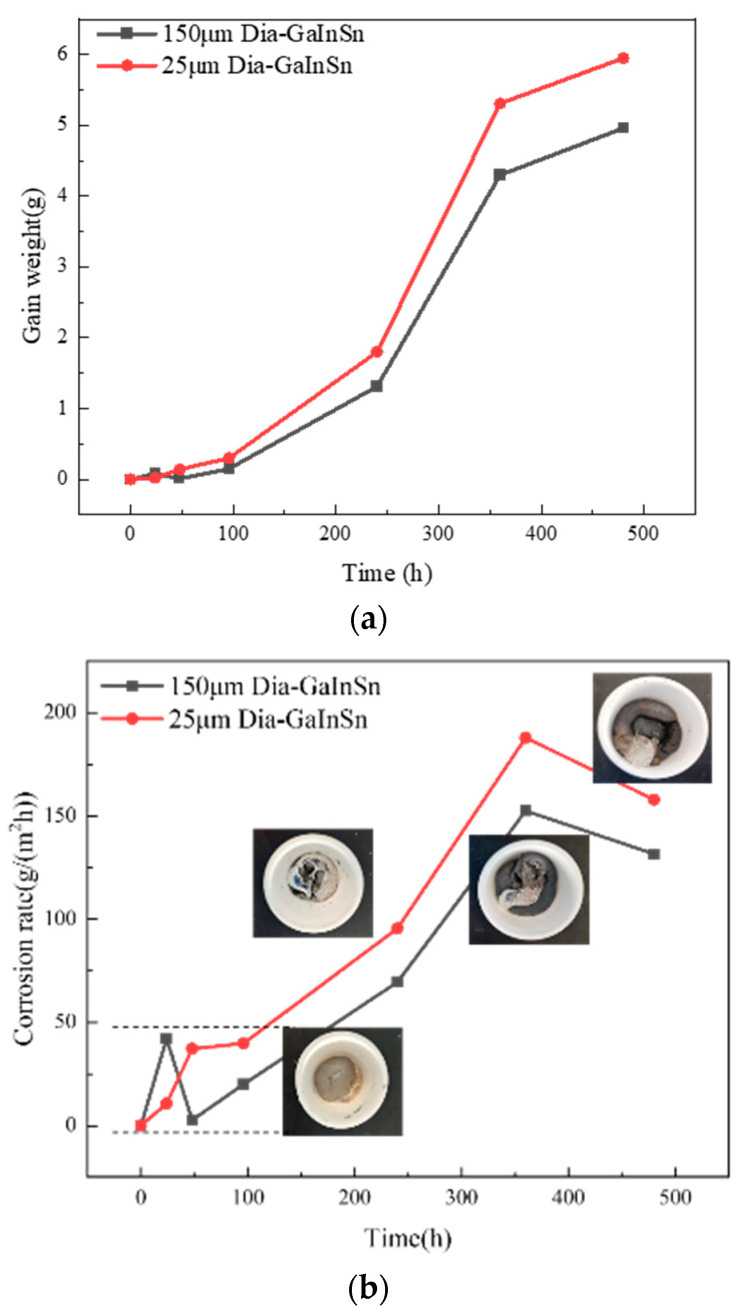
Material corrosion rate diagram. (**a**) Diamond/GaInSn gain weight curve; (**b**) diamond/GaInSn corrosion rate diagram.

**Figure 10 materials-17-01152-f010:**
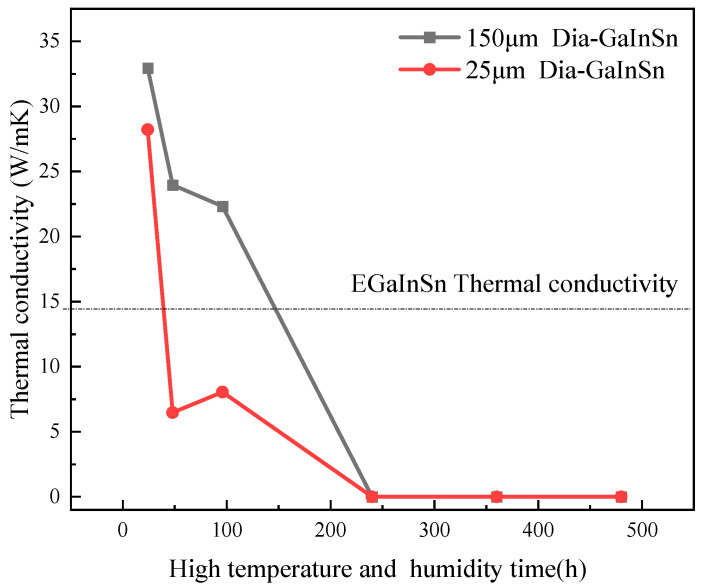
Variation in Dia/GaInSn thermal conductivity under high-temperature and high-humidity conditions.

**Figure 11 materials-17-01152-f011:**
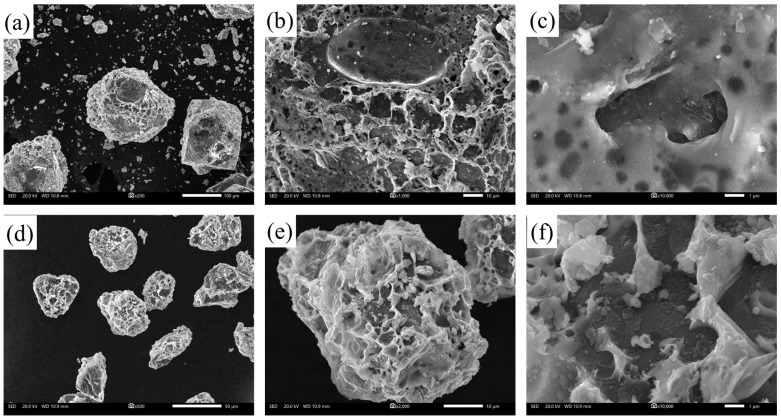
SEM images of the diamond/GaInSn composite material after 240 h of humid heat: (**a**) 150 μm diamond 100×; (**b**) 150 μm diamond 1000×; (**c**) 150 μm diamond 2000×; (**d**) 13 μm diamond 100×; (**e**) 13 μm diamond 1000×; (**f**) 13 μm diamond 2000×.

**Figure 12 materials-17-01152-f012:**
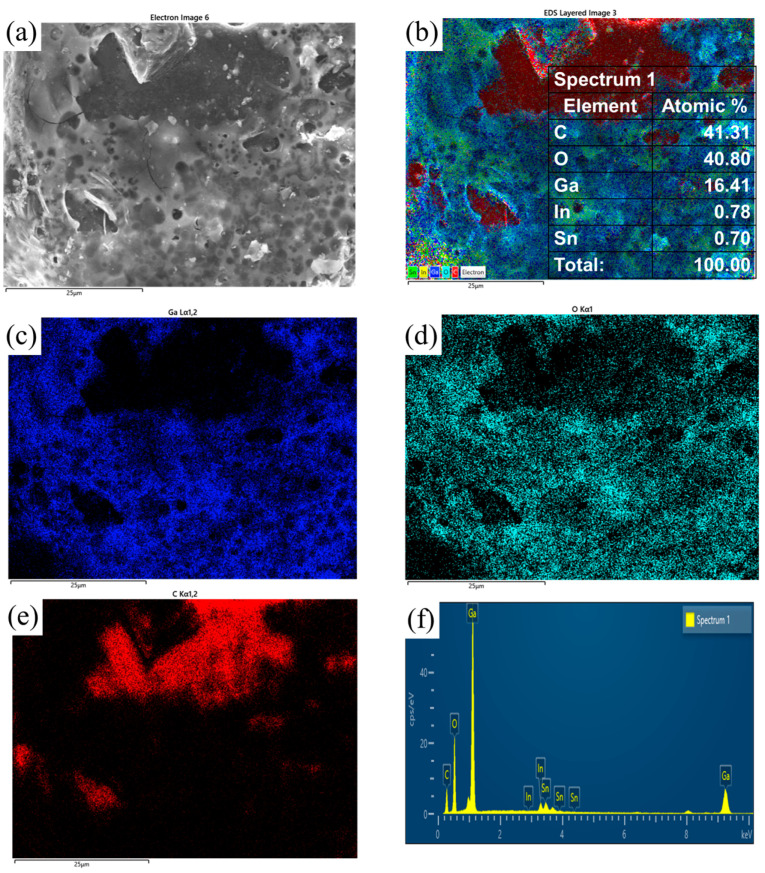
EDS analysis of the diamond surface after 240 h of humid heat. (**a**) SEM image; (**b**) element surface scanning; (**c**) Ga element; (**d**) O element; (**e**) C element; (**f**) elemental analysis.

**Figure 13 materials-17-01152-f013:**
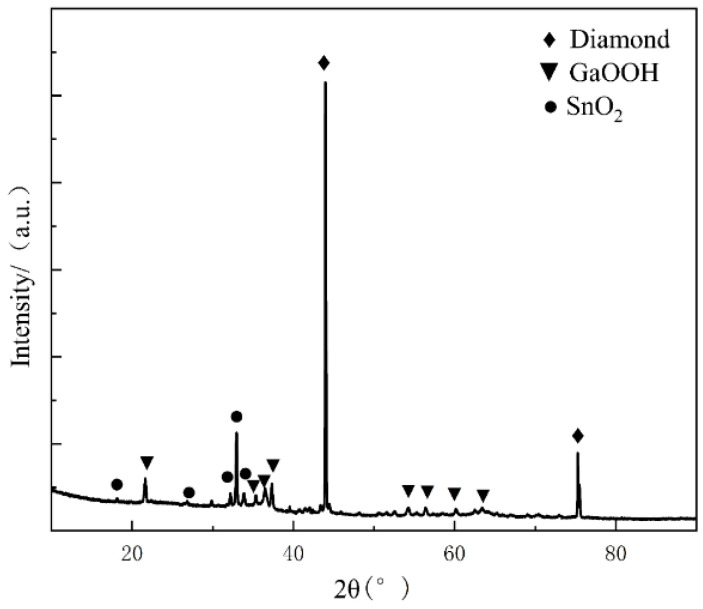
XRD analysis of the diamond surface after 240 h of humid heat.

**Figure 14 materials-17-01152-f014:**
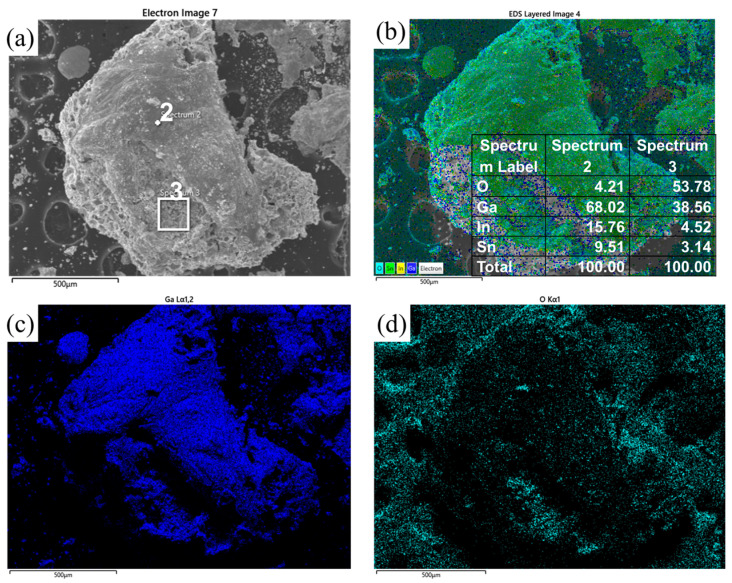
EDS analysis of corrosion products. (**a**) SEM image; (**b**) element surface scanning; (**c**) Ga element; (**d**) O element.

**Figure 15 materials-17-01152-f015:**
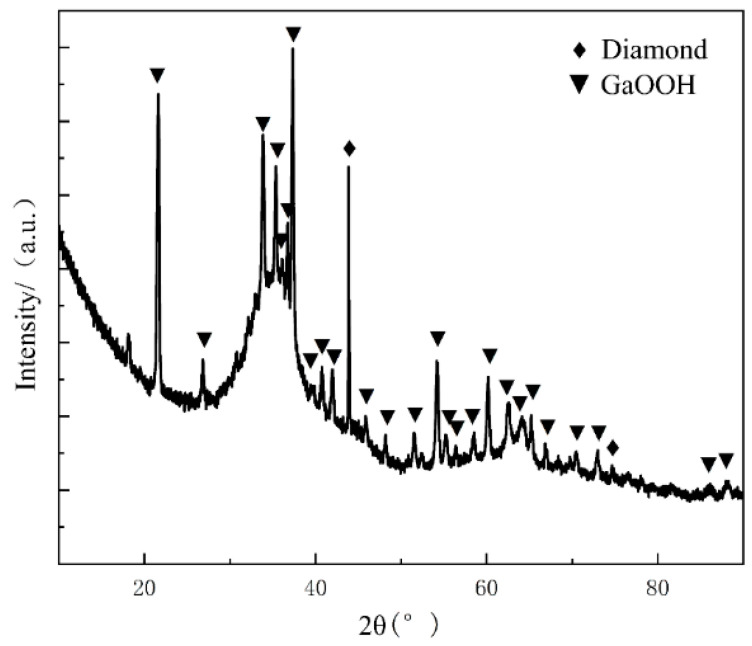
XRD analysis of corrosion products after 240 h of humid heat.

**Figure 16 materials-17-01152-f016:**
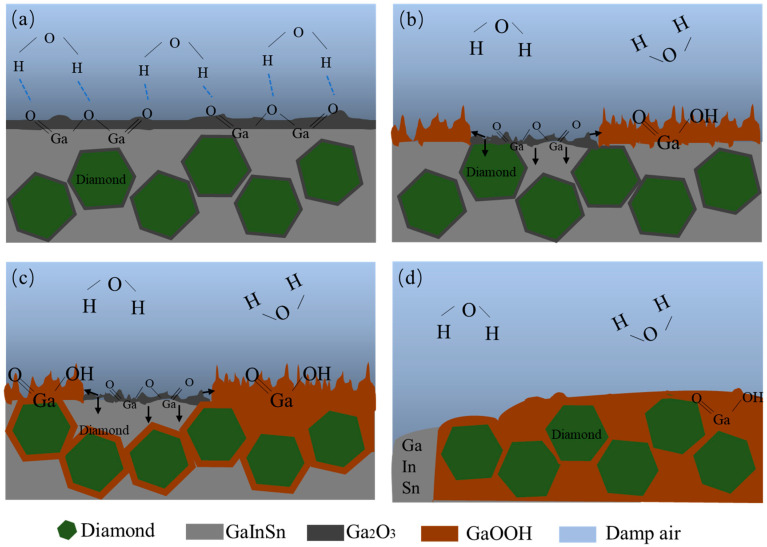
Schematic diagram of the diamond–GaInSn corrosion mechanism at high temperature and high humidity (reaction step: (**a**)–(**b**)–(**c**)–(**d**)).

**Table 1 materials-17-01152-t001:** Properties of composite materials and liquid metals.

Material Property	GaInSn Liquid Metal	GaInSn/Diamond (13 μm)	GaInSn/Diamond (150 μm)
Melting point (°C)	9	9	9
Density (g/cm^3^)	6.17	4.36	4.83
Specific heat (J/gK)	0.38	0.42	0.50
Thermal conductivity (W/mK)	14.94	51.84	70.34

## Data Availability

Data are contained within the article.

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
