# Peer review of "Microstructure and Property Evolution of Diamond/GaInSn Composites under Thermal Load and High Humidity"

_materials, 2024, doi:10.3390/ma17051152_

Round 1

Reviewer 1 Report

Comments and Suggestions for Authors

The authors investigate the practical application of diamond/GaInSn, a composite that finds application in chips used under high temperature, high impact and high humidity environments.

The composite is subjected to high-temperature storage (150oC), high and low-temperature cycling (- 50oC to 125oC) as well as high temperature and high humidity (85oC and 85% humidity) to test its durability during oxidation.

Two grain sizes of diamonds are used in preparing the composites, 25 um and 150 um. It was observed that the porosity of the 25 um diamond composites was higher than that of the 150 um one, leading to easier oxidation reaction spread from the surface into the inner part of the material.

In the high temperature, low temperature cycling, preliminary oxidation promotes wetting leading to an increase in thermal conductivity. The composite then undergoes melting, crystallization, and solidification creating voids, therefore, oxidation that was limited to the surface spread to the inner part of the material, resulting in the thermal conductivity to decrease.

Under high temperature high humidity conditions Ga(OH)2 is generated at the interface between diamond and GaInSn. A honeycomb structure is created leading to reduced wettability between diamond and GaInSn and eventual failure of the material. 

It is concluded that high temperature and high humidity lead to serious failure in the composite therefore these conditions should be avoided in its application.

This is a comprehensive study of the diamond/GaInSn composite. The authors give detailed analysis using SEM (morphology and diamond grain structure analysis), EDS (diamond surface analysis), as well as XRD (diamond surface and corrosion composition analysis).

Where necessary, the results are also presented graphically.

I therefore support the publication of the manuscript.

Comments on the Quality of English Language

The English quality is appropriate, but there is room for improvement.

Author Response

Thank you for reviewing,those comments are very helpful for revising and improving our paper!At the same time, we have also optimized the English expression in the article.

Reviewer 2 Report

Comments and Suggestions for Authors

The work is purely technical in nature. It is not exactly known what the role of diamonds in the mechanism of thermal conduction is. It is known that the thermal conductivity of diamond is at least several dozen times higher than that reported in this paper. Potential heat flow paths are not provided and this requires a deeper explanation. You should also carefully review the works in terms of scientific (technical) language, as there are many jargons and mental abbreviations. The work should be more precise and concise. More precise information on thermal conductivity measurement is also needed.

Below are some of my additional comments

Line 35- insulation – thermal or electrical???

Line 64 - high thermal conductivity, low thermal resistance, - is not the same? High conductivity means low resistance - is not the same? High conductivity means low resistance is it true?

Line 67-68- what mechanism is this about?

Line 75-76 - Two particle sizes of 100 mesh (150 μm) and 500 mesh (13 μm) were used for the diamond?- I think it's about the size of the microcrystallites used in these alloys - this needs to be reworded

Line 80-81- may be in this way? – The  measured the specific performance parameters of liquid metals and composite materials are shown in Table 1

Fig. 1- should be : SEM morphology - SEM photo of morphology

Line 99- with a constant temperature-  at a constant temperature.

Figure 2 Temperature cycle spectrum – this is not the spectrum – Time dependence of temperature cycle

Line 114- 116- what this is about? or about the mass of samples? and then we have the test....after test. This needs to be formulated more concisely and clearly

Line 121- 3D profilometer and Scanning Electron Microscope (SEM) can be  used to characterize the surface morphology and its structure (roughness) and not materials microstructure.

Line 127-the diamond can form a diamond-gallium oxide - what this is about? or a new chemical compound? or is it about covering the surface of the diamond with this compound? this needs to be expressed more clearly

Line 167-170: This is written vaguely. You need to write precisely about the oxidation processes on the surface and inside, and specify exactly what material you are talking about. We have diamond microcrystallites embedded inside the alloy. Deterioration of material properties-which one?,  oxidation inside the material- which material?

How was the oxygen content measured on the surface and inside (core)?

Line 177: wettability is enhanced in morphology -  the sentence is unclear, how can wettability improve morphology? Maybe the other way around?

Comments on the Quality of English Language

The language of work is understandable, but should be more precise and concise

Reviewer 3 Report

Comments and Suggestions for Authors

Manuscript title: Microstructure and property evolution of diamond/GaInSn2 composites under thermal load and high humidity

In this manuscript, Du et al. studied the performance evolution of diamond/GaInSn under high-temperature storage, high-  and low- temperature cycling, and high temperature and high humidity. Two different mesh size of the diamond have been introduced and integrated with GaInSn and studied. Electron microscopic and electrical measurements are performed to studied the structural properties and electrical-thermal properties. The author find that the core oxidation is the key to the degradation of liquid metal composite properties under high temperature storage and high-  and low-temperature cycling conditions. However the data quality, representation and scientific discussion is insufficient for the Journal “Materials” standard.

The written article technically can be improved. The manuscript is lengthy due to the huge number of figures.

The article is not publishable in Materials in the current form.

 Following comments for the author

1.     What is the view of the author to use this materials for chip thermal management. How one can implement with this materials?

2.     how did the author chose the composition. Optimized data presentation is required to validate the best performance claimed.

3.     “Compared with the current commercial TIMs, namely, …………………….reliability is rarely studied” author should include more references including most recent reaearch.

4.     what is the temperature dependent thermal conductivity of bare materials?

5.     ‘the material’s thermal conductivity tends to stabilize after 240–480 h of high-temperature storage” Author should include more data points to find the sharp transition temperature and better understanding of stabilization after 140 K.

6.     In Figure 13 if the author zoom the spectra there is many small diffraction peaks which are not considered in the discussion. Can author verify the unmarked peaks and elaborate the origin of the appearance?

7.     the explanation of the XRD peaks are missing in the discussion. Author must explain the appeared peaks and the corresponding crystallographic planes.

8.     Line 136 ‘sem’ should be upper case.

9.     hard to read the SEM Image scale bars and details are invisible.

10.  There are many typos and subscripts and superscripts typos.

Comments on the Quality of English Language

NA

Round 2

Reviewer 2 Report

Comments and Suggestions for Authors

Thank you very much for taking into account my comments

Comments on the Quality of English Language

Acceptable language

Reviewer 3 Report

Comments and Suggestions for Authors

The authors improve the article. The current form of the article is publishable in the "Materials" without any changes.

Comments on the Quality of English Language

English can be improved.